# Influence of Lipopolysaccharide-Interacting Peptides Fusion with Endolysin LysECD7 and Fatty Acid Derivatization on the Efficacy against *Acinetobacter baumannii* Infection In Vitro and In Vivo

**DOI:** 10.3390/v16050760

**Published:** 2024-05-11

**Authors:** Xiaowan Li, Wenwen Shangguan, Xiaoqian Yang, Xiaoyue Hu, Yanan Li, Wenjie Zhao, Meiqing Feng, Jun Feng

**Affiliations:** 1School of Pharmacy, Fudan University, Shanghai 201203, China; 2Chia Tai Tianqing Pharmaceutical Group Co., Ltd., Nanjing 210046, China; 3China State Institute of Pharmaceutical Industry, Shanghai 201203, China

**Keywords:** endolysin, LysECD7, *Acinetobacter baumannii*, LPS-interacting peptides, fatty acid derivatization, pharmacokinetic

## Abstract

*Acinetobacter baumannii* has developed multiple drug resistances, posing a significant threat to antibiotic efficacy. LysECD7, an endolysin derived from phages, could be a promising therapeutic agent against multi-drug resistance *A. baumannii*. In this study, in order to further enhance the antibacterial efficiency of the engineered LysECD7, a few lipopolysaccharide-interacting peptides (Li5, MSI594 and Li5-MSI) were genetically fused with LysECD7. Based on in vitro antibacterial activity, the fusion protein Lys-Li5-MSI was selected for further modifications aimed at extending its half-life. A cysteine residue was introduced into Lys-Li5-MSI through mutation (Lys-Li5-MSI^V12C^), followed by conjugation with a C16 fatty acid chain via a protonation substitution reaction(V12C-C16). The pharmacokinetic profile of V12C-C16 exhibited a more favorable characteristic in comparison to Lys-Li5-MSI, thereby resulting in enhanced therapeutic efficacy against lethal *A. baumannii* infection in mice. The study provides valuable insights for the development of novel endolysin therapeutics and proposes an alternative therapeutic strategy for combating *A. baumannii* infections.

## 1. Introduction

*Acinetobacter baumannii*, a Gram-negative pathogen, has developed high levels of resistance to a wide range of commonly used antibiotics [1]. Over 80% of *A. baumannii* clinical isolates exhibit multidrug resistance, which poses significant threats to global health [2,3]. Consequently, it is urgent to identify new drugs that can effectively combat antibiotic-resistant bacteria. Bacteriophage endolysins can potentially serve as novel therapeutic agents against antibiotic-resistant *A. baumannii* infections. Endolysins possess the capacity to hydrolyze sugar moieties, peptide backbones, or amide bonds connecting the sugar and peptide moieties of peptidoglycan, resulting in compromised cell wall [4,5,6]. Endolysin LysECD7 is a putative zinc d-alanyl-d-alanine carboxypeptidase (peptidase M15 family) derived from the lytic bacteriophage ECD7. LysECD7 was active against a broad range of Gram-negative clinical isolates, including multidrug-resistant *A. baumannii*. Importantly, LysECD7 did not contribute to the development of short-term resistance [7,8]. Moreover, it demonstrated efficacy in animal models of wound and burn skin infections through epicutaneous administration [9]. However, the inhibitory effect of LysECD7 on bacterial growth activity was limited with a reported minimum inhibitory concentration (MIC) exceeding 1 mg/mL [10]. To date, LysECD7 has not been employed for the treatment of systemic infection in animal models.

To enhance efficiency against Gram-negative bacteria, a few antimicrobial peptides have been genetically fused with native endolysin [11]. Most peptides comprise cationic and amphipathic amino acids that can interact with negatively charged lipopolysaccharide (LPS). This interaction destabilizes the LPS structure, enabling the endolysin to access peptidoglycans. For example, the fusion of sheep myeloid peptide 29 (SMAP29) with endolysin KZ144, known as Art175, exhibited remarkable efficiency against multidrug-resistant stationary-phase cells of *A. baumannii* and their persisters [2,12,13]. The truncated form of SMAP29 was further fused with the endolysins LysECD7 and L-KPP10, thereby enhancing their antibacterial activities [10]. Additionally, MSI-594 and Li5 are also potential peptides that interact with LPS. MSI-594 has been demonstrated to possess the ability to bind LPS and disrupt LPS aggregates into smaller-sized particles [14,15]. Moreover, the Li5 peptide screened using the phage display method showed a high affinity for LPS (10^−7^–10^−8^ mol/L) [16]. However, their potential for fusion with endolysins to enhance antibacterial activities against Gram-negative bacteria remains unexplored. 

Despite demonstrating efficacy in vitro, there are still limitations for endolysin as potential therapeutics, particularly in the treatment of systemic infections [17]. They generally exhibit a short serum circulation half-life, with the currently reported half-life of natural endolysins being less than 1 h [18,19,20]. Several strategies have been proposed to extend the half-life of endolysins, including conjugation with poly-ethylene glycol (PEG), albumin-binding domain (ABD) and dimerization. The half-lives of Lysostaphin and Cpl-1 have been prolonged through conjugation with PEG [19,21], due to an enhanced water-binding capacity, increased particle size, and hindered renal filtration [22]. Dimerization of Cpl-1 and Lysostaphin resulted in an increase in molecular size that surpassed the renal filtration cut-off value (40–50 kDa), thereby extending their half-lives by 2 to 10-fold [23,24]. For endolysin M23 and CH-GH15, fusion with ABD extended their half-lives by 1.5 to 2-fold [17]. Although these strategies were efficient in extending the half-lives of endolysins, in vitro antibacterial activities of most endolysins were compromised by structural changes [17,21,24]. Another promising approach for prolonging the half-life involves fatty acid derivatization. Products derived from fatty acids can form a high-affinity complex with human serum albumin (HSA), which is then protected from renal filtration due to its large hydrodynamic volume and preserved from lysosomal degradation through recycling by the neonatal Fc receptor (FcRn). This strategy has been successfully employed to extend the half-lives of peptides such as insulin, and derived peptides had comparable activities to un-derived peptides [25]. However, to date, there have been no reports on the application of fatty acid derivatization to endolysins.

This study investigated the impact of fusing various LPS-interacting peptides including MSI-594, Li5 and Li5-MSI with LysECD7 on activity. In addition, we studied the influences of fatty acid derivatization of endolysin on pharmacodynamics and treatment efficacy in mice with peritoneal infections.

## 2. Materials and Methods

### 2.1. Recombinant Fusion Proteins Design

To investigate the influence of distinct LPS-interacting peptides on LysECD7, we generated Lys-Li5, Lys-MSI, and Lys-Li5-MSI constructs by separately fusing the Li5, MSI594, and Li5-MSI peptides (Appendix A) to the C-terminus of LysECD7 using a GGGGS repeat sequence as a linker. 

To react with the fatty acid side chain effectively, a free cysteine was introduced to Lys-Li5-MSI. The predicted three-dimensional structure of LysECD7 was established and analyzed using the Swiss-model (https://swissmodel.expasy.org, (accessed on 8 July 2023)). The 12th amino acid valine, positioned on the protein surface and distant from the active domain, was substituted with a free cysteine (Appendix A).

All fusion proteins mentioned, including the comparison protein LysECD7, featured an 8-His tag in their C-terminal region. 

### 2.2. Plasmid Construction and Bacterial Strains

The coding sequences of recombinant proteins were chemically synthesized by Sangon Biotech Co., Ltd. (Shanghai, China). These sequences were inserted into the NdeI and BamHI sites of the pET29a vector. The resulting plasmids were electro-transformed into *E. coli* BL21(DE3) cells for protein production. The *A. baumannii* ATCC 19606 strain was obtained from the American Type Culture Collection (Manassas, VA, USA).

### 2.3. Protein Production and Purification 

The engineered *E. coli* BL21 strains were inoculated in LB medium supplemented with 25 μg/mL kanamycin and incubated to the exponential phase (OD_600nm_ = 0.8–1.2) in a shaker (37 °C, 220 rpm). The cultured bacteria were subsequently transferred to a fermentation medium (glucose 2.0 g; yeast extract 5.0 g; glycerin 20.0 g; Na_2_HPO_4_ 17.2 g; KH_2_PO_4_ 3.0 g; NaCl 0.5 g; (NH_4_)_2_SO_4_ 3.0 g in a 1.0-L tap water) at a volume ratio of 3%. *E. coli* was cultured to an OD_600nm_ of 1.0–2.0, then induced by isopropyl β-D-1-thiogalactopyranoside and grown overnight at 37 °C. The bacterial cells were harvested and lysed using a high-pressure homogenizer (800 bar). 

For the recombinant proteins LysECD7, Lys-Li5, and Lys-MSI were mainly produced in a soluble form, the supernatant was purified after passing through a 0.22 μm filter. However, for Lys-Li5-MSI and Lys-Li5-MSI ^V12C^ predominantly existed as inclusion bodies and denaturation and renaturation were required prior to purification. Consequently, 6 mol/L guanidine hydrochloride (pH 11.0) was used to dissolve the inclusion bodies. The denatured proteins were diluted 10-fold in renaturation buffer (2 mol/L urea, 0.5 mol/L arginine, 25 mmol/L Tris-HCl, pH 8.0). 

Proteins were purified using a Ni-NTA affinity column (Cytiva Corporation, Marlborough, MA, USA). Following the pooling of samples, the column was washed with solution A (50 mmol/L imidazole, 250 mmol/L NaCl, 25 mmol/L Tris-HCl, pH 8.0), and the His-tagged proteins were eluted with solution B (500 mmol/L imidazole, 250 mmol/L NaCl, 25 mmol/L Tris-HCl, pH 8.0). Hitrap SPHP (Cytiva Corporation, USA) was used for further purification, and gradient elution was performed with 1 mol/L NaCl, 25 mmol/L MES, pH 6.5. The sample buffer was replaced with Tris-HCl or phosphate buffer using an Amicon Ultra-15 3 K filtration system (Sigma-Aldrich, Darmstadt, Germany).

Protein purity was analyzed on 15% SDS-PAGE gels and quantified using BCA kits (Shanghai Epizyme Biomedical Technology Co., Ltd., Shanghai, China). 

### 2.4. Bactericidal Activity Assay

The *A. baumannii* ATCC19606 strain was inoculated in 30 mL of LB medium and cultured at 37 °C to the exponential phase (OD_600nm_ ≈ 1.0). Subsequently, the cultured bacteria were centrifuged at 5000× *g* for 5 min. The supernatant was discarded, and an equal volume of 20 mmol/L Tris-HCl (pH 7.5) was added to suspend the bacteria. The bacterial suspension was diluted 100-fold with Tris-HCl buffer to achieve a bacterial count of 10^5^–10^6^ CFU/mL.

The protein concentration was respectively diluted to 64, 32, 16, 8, and 4 μg/mL. The diluted proteins were mixed with the bacterial solution at a ratio of 1:1, with a buffer without protein as the negative control. The mixtures were incubated at 37 °C for 30 min with shaking at 200 rpm. The reaction solutions were diluted 10-fold with Tris-HCl buffer and 100 μL was plated on LB solid medium. The plates were incubated overnight at 37 °C, and the single colonies that grew out were counted. The experiments were conducted in triplicates.

The activities were calculated using the following formula:

Bactericidal activity (%) = 100% − (CFU_exp_/CFU_cont_) × 100%, where CFU_exp_ represents the number of single colonies grown on the experimental culture plates and CFU_cont_ refers to the number of single colonies on the control culture plates. Bactericidal activity was considered meaningful when it exceeded 33 [9,10].

### 2.5. Determination of Minimum Inhibitory Concentration

The MIC was determined using the broth micro-dilution method according to the Clinical and Laboratory Standards Institute guidelines [26]. A mid-logarithmic growth phase culture of *A. baumannii* ATCC19606 was diluted to 1 × 10^6^ CFU/mL using MHB. Bacteria were exposed to LysECD7 and fusion proteins. The final concentrations of the proteins were in the range of 1–1024 μg/mL. Bacteria in MHB without proteins were used as controls. The volume of the solution in each well was 200 μL. After incubation for 16–20 h at 37 °C, the lowest protein concentration with no visible growth was considered the MIC. 

### 2.6. Fatty Acid Derivatization

The cysteine mutant of Lys-Li5-MSI (Lys-Li5-MSI ^V12C^), which was present as inclusion bodies, was dissolved in 6 mol/L guanidine hydrochloride, and 5 mmol/L TCEP was used to disrupt the dimer. The solution was then diluted 10-fold with renaturation buffer. The renatured protein was purified using a Ni-NTA affinity column and eluted with 500 mmol/L imidazole. The sample buffer was then replaced with PBS by filtration.

Purified Lys-Li5-MSI ^V12C^ protein (1 mg/mL) was treated with 1 mmol/L TCEP and incubated for 1 h. The C16 fatty acid side chain (HO_2_C-(CH_2_)_14_-CO-γGlu-AEEA-AEEA-Lys (ε-Acetyl bromide)-OH (synthesized in our laboratory) was solubilized by anhydrous ethanol and reacted with Lys-Li5-MSI ^V12C^ at a ratio of 5:1 for 1 h. 

The Lys-Li5-MSI ^V12C^ protein and lipidation product, V12C-C16, were analyzed using HPLC. The conditions were as follows: Column, YMC Triart C4 (5 μm, 4.6 mm × 250 mm); solution A, 0.1% TFA-H_2_O; solution B, 0.1% TFA-ACN; gradient, 30%, 4 min; 30–45%, 15 min; 45–80%, 2 min. The unreacted Lys-Li5-MSI ^V12C^ protein and lipidation product V12C-C16 were separated by reverse-phase purification using a Kromasil 300-10-C4 column (10 mm × 250 mm). The gradient conditions were 20%, 4 min; 20–30%, 8 min; 30–50%, 60 min; 50–80%, 8 min. The purified product was mixed and rotary evaporated to remove ACN. The sample buffer was then replaced with PBS by filtration. The molecular weight of V12C-C16 was determined by electrospray ionization mass spectrometry.

The MIC values of Lys-Li5-MSI ^V12C^ and V12C-C16 were determined in MHB and MHB supplement with 1% human serum albumin.

### 2.7. ELISA

The concentrations of Lys-Li5-MSI and V12C-C16 were determined by sandwich ELISA. Briefly, the 96-well plates (Thermo Fisher Scientific™, Waltham, MA, USA) were coated with LPS from *Escherichia coli* O55:B5 (Sigma Aldrich, Saint Louis, MO, USA) and incubated for 2 h at 37 °C. The plate was washed with PBST and blocked with 5% non-fat milk for 1 h. The diluted samples were then added, and the plate was incubated for 1 h. The plate was then washed, and HRP-conjugated rabbit polyclonal anti-LysECD7 (Anhui Gene Universal Technology Co., Ltd., Chuzhou, China) was added. After incubation for 40 min, the plate was washed and incubated with TMB peroxidase substrate (Thermo Fisher Scientific™, USA) for 20 min. The reaction was stopped with sulfuric acid, and the optical density was measured at a wavelength of 450 nm.

### 2.8. Pharmacokinetic Studies

Male ICR (Institute of Cancer Research) mice (26–27 g) were divided into two groups and intravenously administered a single dose of Lys-Li5-MSI and V12C-C16 (15 mg/kg). Blood samples were collected at 5, 15, and 30 min, and at 1, 2, 4, 6, and 8 h after retro-orbital bleeding (n = 3 animals per time point). Blood samples were collected in EDTA-coated test tubes and were stored on ice until centrifugation at 1200× *g* for 10 min at 4 °C. Plasma was transferred to a microcentrifuge tube and stored at −80 °C until analysis. The plasma samples were diluted and their concentrations were assayed using the sandwich ELISA method. Pharmacokinetic parameters were calculated using a two-compartment model with Drug and Statistics software (DAS, ver. 2.0; Mathematical Pharmacology Professional Committee of China, Shanghai, China).

### 2.9. In Vivo Efficacy Evaluation

*A. baumannii* ATCC19606 was used to induce an intraperitoneal infection model. A preliminary experiment was conducted to determine the minimum lethal dose of the strains (3.33 × 10^8^ CFU/mL). A total of 52 BALB/c mice (17–22 g) were randomly assigned to four groups, with each group consisting of 13 mice. One group served as the uninfected control. The other three group mice were intraperitoneally injected with 500 μL 3.3 × 10^8^ CFU/mL of *A. baumannii* ATCC19606. After 0.5 h, the mice were administered 50 mg/kg Lys-Li5-MSI, 50 mg/kg V12C-C16 and PBS via intravenous injection. After 7 h, 3 mice were taken from each group. The mouse liver, spleen, lungs, and kidneys were harvested and weighed. PBS was added for tissue homogenization in a ratio of 0.1 g tissue/0.9 mL PBS. After dilution, 10 mL molten MHB agar medium was added. After solidification, the medium was incubated at 37 °C for 18–24 h before counting the bacterial load. The survival of the remaining ten mice in each group was monitored daily for a period of 6 days.

## 3. Results

### 3.1. Production of Fusion Proteins and Determination of Antibacterial Efficiency In Vitro

The endolysin LysECD7 was investigated for potential enhancement of its bacterial activity through fusion with peptides that bind to LPS. We fused MSI594, Li5, and Li5-MSI594 (Appendix A) to the C-terminus of LysECD7 using a GS linker, resulting in chimeras named Lys-MSI, Lys-Li5 and Lys-Li5-MSI. The recombinant proteins were produced via fermentation. Subsequently, these proteins were pretreated and purified using Ni-NTA affinity and cation-exchange chromatography. The resulting purified proteins were analyzed by 15% SDS-PAGE (Figure 1a). The observed protein bands corresponded well with their theoretical molecular masses of 15.8 K (lane1 LysECD7), 17.9 K (lane2 Lys-Li5), 18.9 K (lane3 Lys-MSI), and 21.2 K (lane4 Lys-Li5-MSI) respectively. All the purified proteins exhibited purities exceeding 90%. 

The bactericidal activities of LysECD7 and its fusion proteins against exponentially growing bacterial cells were shown to be concentration-dependent. The minimal active concentration (bactericidal activity > 33%) after 30 min of incubation in 20 mmol/L Tris HCl buffer (pH 7.5) was observed to be 16 μg/mL for LysECD7 and Lys-Li5, 8 μg/mL for Lys-MSI, and 4 μg/mL for Lys-Li5-MSI (Figure 1b). Additionally, the inhibitory effect on bacterial growth was assessed. The MIC of LysECD7 exceeded 1024 μg/mL, which was consistent with previous reports [10]. Lys-Li5 exhibited similar results. However, the MICs of Lys-MSI and Lys-Li5-MSI were identical at 16 μg/mL (Table 1). These results suggested that the fusion of MSI594 and Li5-MSI resulted in enhanced antimicrobial activities compared to LysECD7. However, the efficacy of fusion Li5 peptide was limited.

### 3.2. Fatty Acid Derivatization

To extend the serum circulating half-life, the fatty acid derivatization technology was utilized. Given the absence of cysteine in Lys-Li5-MSI, the introduction of free cysteine into the sequence represents a suitable strategy for site-directed derivatization. We introduced cysteine at the 12th amino acid residue via substitution of valine. This substitution, being at the surface of the protein structure and distant from the active site (Appendix A), might not interfere with the activity. The mutated protein, Lys-Li5-MSI ^V12C^, was obtained following renaturation and purification. The MICs of both Lys-Li5-MSI and Lys-Li5-MSI ^V12C^ were identical, at 16 μg/mL (Table 1). This finding confirmed that the introduction of cysteine did not affect the antibacterial activity. 

The free thiol group in Lys-Li5-MSI ^V12C^ could react with the acetyl bromide group in the C16 fatty acid side chain (HO_2_C-(CH_2_)_14_-CO-γGlu-AEEA-AEEA-Lys (ε-Acetyl bromide)-OH) via a protonation substitution reaction, resulting in the product V12C-C16 (Figure 2a,b). The mass of the product was confirmed (Appendix A). The two compounds, Lys-Li5-MSI ^V12C^ and V12C-C16, exhibited differences in hydrophobicity and were used for further analysis and purification (Figure 2b). The MICs of the purified Lys-Li5-MSI ^V12C^ and V12C-C16 were identical at 16 μg/mL (Table 1). In presence of HSA, there was no change in the MIC value. These results demonstrate that fatty acid derivatization and incubation with HSA did not affect the antibacterial activity of the fusion protein.

### 3.3. Pharmacokinetics

To compare the half-lives of Lys-Li5-MSI and V12C-C16, we conducted pharmacokinetic experiments in vivo. ICR mice were intravenously administered a dose of 15 mg/kg for both Lys-Li5-MSI and V12C-C16. The plasma concentrations were determined using sandwich ELISA, and the results are presented in Figure 3. In vivo, Lys-Li5-MSI exhibited rapid elimination, with a concentration of <1 μg/mL at 2 h, whereas V12C-C16 showed a slower elimination rate with detectable concentration (>1 μg/mL) even at 6 h. The pharmacokinetic parameters are listed in Table 2. After intravenous administration, the half-life of V12C-C16 was observed to be extended by 3.9-fold compared to Lys-Li5-MSI. In addition, the plasma clearance rate of V12C-C16 was reduced 2.5-fold slower than that of Lys-Li5-MSI. Furthermore, the total exposure to the circulating drug or the area under the curve of V12C-C16 was approximately 2.3-fold higher than that of Lys-Li5-MSI. These studies suggested that following intravenous administration, V12C-C16 exhibited a more favorable pharmacokinetic profile than Lys-Li5-MSI. 

### 3.4. In Vivo Efficacy in a Murine Model

Finally, we investigated whether fatty acid derivatization would result in enhanced therapeutic efficacy. The in vivo antibacterial efficiencies of Lys-Li5-MSI and V12C-C16 was evaluated using a murine model of intraperitoneal infection. They were selected based on their demonstrated high antimicrobial activity in vitro. In this study, mice were infected with *A. baumannii* ATCC19606 through peritoneal injection. After 0.5 h, mice were treated with an intravenous injection of Lys-Li5-MSI, V12C-C16 or PBS solution. After 7 h, there were no significant differences between Lys-Li5-MSI and PBS groups regarding bacterial load in different issues. However, the bacterial load in the liver and lung was reduced in the V12C-C16 treatment groups compared to both PBS and Lys-Li5-MSI group (Figure 4). The survival rates of uninfected mice and infected mice treated with various samples (n = 10 per group) were monitored for a duration of 6 days, as depicted in Figure 5. Only 10% mice survived throughout the experiment in both the Lys-Li5-MSI and PBS group. But in the V12C-C16 treatment group, the survival rate was 40%. These results suggested that compared to Lys-Li5-MSI, its fatty acid derivative V12C-C16 exhibited superior therapeutic efficacy against *A. baumannii* infection in mice.

## 4. Discussion

The endolysin LysECD7 was fused with three different LPS-interacting peptides (Li5, MSI and Li5-MSI). The fusion proteins Lys-MSI and Lys-Li5-MSI exerted 2-4-fold increased bactericidal activities and significantly improved inhibitory effects in vitro. Both fusion proteins contained MSI-594 sequence, which is known to disrupt LPS [14]. The enhanced antibacterial efficacy of Lys-MSI and Lys-Li5-MSI may be attributed to the MSI594-assisted uptake of the endolysin moiety through the outer membrane, followed by enzymatic degradation of the PG layer and osmotic lysis. This mechanism has been studied previously studied for Art175 (a fusion protein of endolysin KZ144 and peptide SMAP29). In the case of bacteria treated with Art175, complete lysis was mostly preceded by cytoplasmic membrane bulging through a hole created in the cell wall [12]. However, treatment with SMAP29 only induced surface morphological changes, the intact PG sacculi remained and no cell lysis was observed. The action of Art-175 exhibited greater rapidity compared to SMAP29 [12]. These findings substantiate the synergistic effect of KZ144 and SMAP29. Given that both MSI594 and SMAP29 were cationic antibacterial peptides and disrupt LPS, it seems reasonable to assume that Lys-MSI and Lys-Li5-MSI adopt a similar mechanism to improve their antibacterial efficacy. In contrast to their effects, fusion with the Li5 peptide did not enhance antibacterial efficacy, potentially due to Li5 peptide target LPS by affinity interaction [9], which does not disrupt LPS, and consequently fails to transport the endolysin through the outer membrane of Gram-negative bacteria. 

Fatty acid derivatization did not compromise the antimicrobial activity of Lys-Li5-MSI in vitro, even in the presence of human serum albumin. This could be attributed to the targeted modification site being located on the protein surface and distant from the active domain (Appendix A). Furthermore, the presence of a linker within the fatty acid side chain (-γGlu-AEEA-AEEA) may attenuate the influence of fatty acid-albumin complexes on enzymatic cleavage of the bacterial peptidoglycan. In previous studies, PEGylation, ABD fusion and dimerization have been shown to improve the half-lives of endolysins. However, in most cases, these structural changes on endolysins also led to a significant decrease in activity. The conjugation of PEG chains significantly decreased activity of lysostaphin [19] and Cpl-1 [21]. Despite adjusting the linker length and position of the ABD within the PGH as done by Anna M et al. [17], a reduction in activity was still observed for all ABD fusions compared to their parental enzymes in in vitro assays. Dimerization proved suitable for endolysins that tend to form multimers [23]; however, for other endolysins, like lysostaphin, dimerization may lead to a substantial decrease in activity [24]. Compared to these approaches, fatty acid derivatization appeared to be superior in terms of its effect on endolysin activity.

Fatty acid derivative V12C-C16 demonstrated superior efficacy in vivo compared to Lys-Li5-MSI, despite their equivalent activities observed in vitro. The in vivo efficacy of Lys-Li5-MSI was completely abolished, failing to reduce bacterial load in tissues and rescue infected mice from mortality. One potential contributing factor may be the rapid clearance of Lys-Li5-MSI from the bloodstream following a single intravenous dose. The half-life of Lys-Li5-MSI was short (0.397 h), which might not provide sufficient time for maintaining effective therapeutic concentrations. After fatty acid derivatization, the serum circulating half-life of V12C-C16 was extended by 3.9-fold, resulting in a prolonged duration for maintaining an effective concentration in the bloodstream. This enhancement may potentially lead to improved treatment efficacy. Previous studies demonstrated that fusion of endolysin M23 and ABD led to an increase in plasma half-life, thereby enhancing efficacy in treating *S. aureus* bacteremia in vivo [17]. These findings are consistent with our results. However, the possibility cannot be excluded that other factors besides half-life extension contributed to the observed outcome of the mouse study, including the stability of Lys-Li5-MSI and V12C-C16 in murine system, as well as potential inhibition of enzyme activity under physical condition. Further studies are required to investigate the possible roles of these factors on the efficacy.

In conclusion, we have demonstrated that fusion of LPS-interacting peptides with endolysin LysECD7 could enhance the in vitro antibacterial activity against *A. baumannii*. Furthermore, fatty acid derivatization extends the half-life in vivo, thereby improving treatment efficacy in a murine model of intraperitoneal infection. 

## Figures and Tables

**Figure 1 viruses-16-00760-f001:**
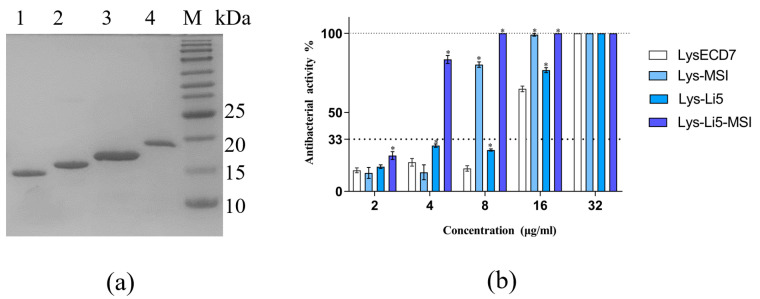
Production and antibacterial activities analysis of LysECD7 and its LPS-interaction peptides fusion proteins. (**a**) Purified fusion proteins analyzed by 15% SDS-PAGE (lane1-M: 1. LysECD7; 2. Lys-Li5; 3. Lys-MSI; 4. Lys-Li5-MSI; M. Marker); (**b**) The antibacterial activities of LysECD7, Lys-MSI, Lys-Li5 and Lys-Li5-MSI. The mean values are shown from three independent experiments ± SD. An asterisk (*) indicates a significant difference in bactericidal activities compared to LysECD7 (*p* < 0.05, two-way ANOVA with Dunnett’s multiple comparisons test).

**Figure 2 viruses-16-00760-f002:**
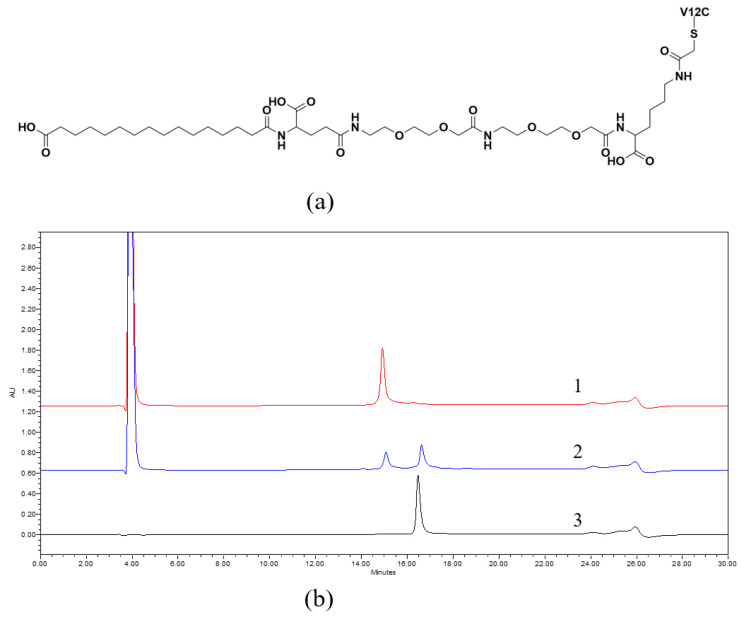
Fatty acid derivatization. (**a**) The structure of the fatty acid derivative V12C-C16. The V12C represents the Lys-Li5-MSI ^V12C^ protein in which 12th amino acid valine was cysteine. (**b**) The HPLC analysis of fatty acid derivative reaction. Curve 1: Lys-Li5-MSI ^V12C^ protein, Curve 2: the outcome of fatty acid derivatization reaction. Curve 3: purified V12C-C16 product.

**Figure 3 viruses-16-00760-f003:**
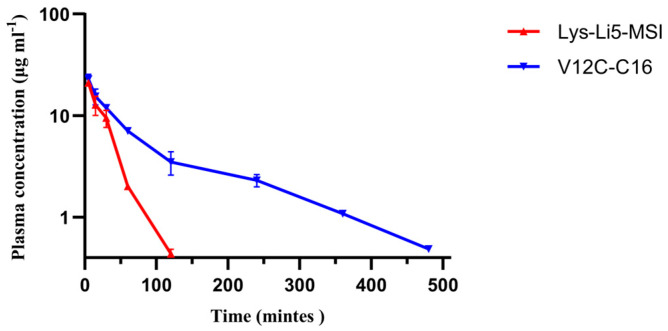
Comparison of the pharmacokinetic profiles of Lys-Li5-MSI and V12C-C16 after intravenous administration in mice. Data are presented as means ± SD (n = 3 per group).

**Figure 4 viruses-16-00760-f004:**
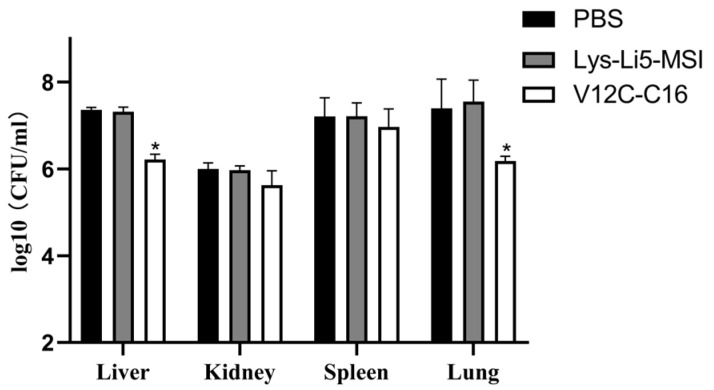
The bacterial load of different tissues of infected mice. Data are presented as means ± SD (n = 3 per group). An asterisk (*) indicates a significant difference compared to PBS group (*p* < 0.05).

**Figure 5 viruses-16-00760-f005:**
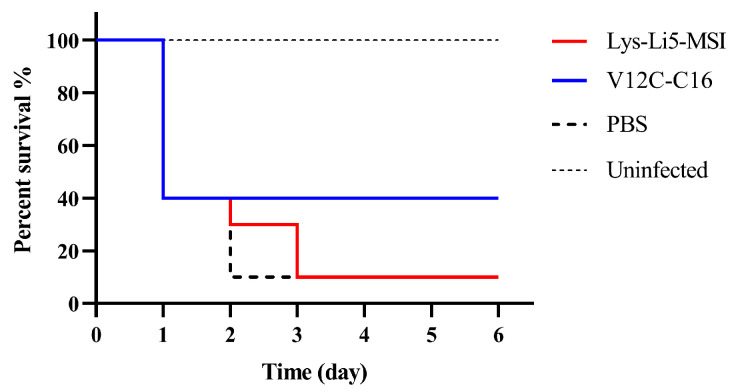
Survival rate of mice. n = 10 per group.

**Table 1 viruses-16-00760-t001:** The MICs of LysECD7 and fusion proteins.

Sample	MIC (μg/mL)
LysECD7	>1024
Lys-Li5	>1024
Lys-MSI	16
Lys-Li5-MSI	16
Lys-Li5-MSI ^V12C^	16
V12C-C16	16

**Table 2 viruses-16-00760-t002:** Pharmacokinetic parameters of Lys-Li5-MSI and V12C-C16 following i.v. administration in mice.

Sample	Half-Life (h)	Vd (L kg^−1^)	CL (L h^−1^ kg^−1^)	AUC (0–∞) (μg L^−1^ h^−1^)
Lys-Li5-MSI	0.397	0.021	1.034	14,512
V12C-C16	1.544	0.452	0.411	32,754

Abbreviations: Vd, volume of distribution; CL, clearance; AUC, area under the curve.

## Data Availability

Data are contained within the article.

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
