# Peer review of "Influence of Lipopolysaccharide-Interacting Peptides Fusion with Endolysin LysECD7 and Fatty Acid Derivatization on the Efficacy against Acinetobacter baumannii Infection In Vitro and In Vivo"

_viruses, 2024, doi:10.3390/v16050760_

Round 1

Reviewer 1 Report

Comments and Suggestions for Authors

Imminent loss of antibiotics due to increasing prevalence of drug resistant bacteria is a major threat to human well-being. This is why there is a search for alternative antimicrobials, including phages and phage-derived molecules. Bacteriophages targeting Acinetobacter baumannii have some of the narrowest known host ranges, making it a particularly worrisome pathogen. Unlike bacteriophages themselves, phage endolysins don't rely on host cell molecular machinery and generally have broader activity spectra. Allegedly, bacteria fail to develop endolysin resistance. The weak sides of endolysins include the lack of self-replication and high decay rates.

By preparing fusion constructs of a previously described endolysin with lipopolysaccharide-interacting peptides, the Authors increased its in vitro antibacterial activity. Further modification of one most active protein with a fatty acid residue significantly improved its pharmacokinetic profile.

Introduction provides necessary information, but contains a lot of details more suitable for Discussion. The section Materials and Methods describes the design of three fusion constructs of the endolysin with lipopolysaccharide-interacting peptides and the related procedures aimed at obtaining protein preparations. The following subsections deal with determination of antibacterial activity, lipidation, a pharmacokinetic experiment and eradication of A. baumannii infection in animal model. The employed experimental approaches seem to be relevant to the aims of the Study. The Results are presented fairly well. In Discussion, the language is vague. At the beginning one would expect one or several general phrases. There is a re-iteration of the same idea on the Lines 351 and 361. Nevertheless, the section provides an insight into possible molecular mechanisms underlying observed phenomena and mentions related studies.

One necessary detail apparently missing from the Discussion is an honest comparison of V12C-C16 therapeutic potential to the efficacy of other phage endolysins. In my perception, it is very modest. Also, a brief note on this would be welcome in the Abstract.

All in all, the study appears to be convincing and mature, though it is not without drawbacks.

Minor comments

Introduction and Discussion are poorly structured. Some way to improve writing style should be found.

Consider renaming Lys-Li5-MSI and/or V12C. The two proteins differ by one amino acid residue and the prominent difference between the names significantly complicates reading.

Line 168, “The V12C inclusion body”. Please clarify. At the start of the subsection, I would expect mentioning Lys-Li5-MSI.

Lines 228-229. “The recombinant proteins were produced via fermentation”. Check if fermentation is an appropriate word.

Figure 5 is not referenced in the text. To my mind, "untreated" means "infected, but not cured". Please check and add an informative caption.

Data availability statement claims that no new data were created or analyzed in the study. I believe, there is a kind of mistake, since the study is original, as opposed to literature review.

Other thoughts

Outbred mice are genetically diverse. Therefore, the use of these animals in experiments increases noise in the data.

Comments on the Quality of English Language

/

Reviewer 2 Report

Comments and Suggestions for Authors

This is a nice contribution on the potential use of the phage endolysin LysECD7 to control infections by A. baumannii. I enjoyed the good quality of the data and the logical flow of the paper. There are a few minor issues that should be corrected.

1. line 82: spell out "FcRn recycling" at its first mention.

2. line 281: spell out "ICR mice" at its first mention.

3. line 356: correct to: "half-life extension contributed to..."

4. line 376: "Anna M et al." is not listed in the reference list, maybe as (17)?

Reviewer 3 Report

Comments and Suggestions for Authors

The research article by Li et al., on "Influence of lipopolysaccharide-interacting peptides fusion with endolysin LysECD7 and fatty acid derivatization on the efficacy against Acinetobacter baumannii infection in vitro and in vivo" describes the antibacterial potential of chimeric endolysin, LysECD7. The manuscript is well presented and the methods are scientifically sound as well as the interpretation/ discussion. The evaluation of antibacterial activity in mice models is interesting which is also a key topic of the study. 

Minor:

1. Line no. 19: Correction: multi-drug resistance

2. Line no. 210: More details are needed for in vivo study. Why ICR mice? why 0.5 h after infection and before administration, what will happen if it will be more than 2 hrs? 

3. Line no. 310-315: The survival rate is much lesser than expected especially from in vitro studies and half-life. 
